# Exploring Relationships among Grapevine Chemical and Physiological Parameters and Leaf and Berry Mycobiome Composition

**DOI:** 10.3390/plants11151924

**Published:** 2022-07-25

**Authors:** Anna Molnár, József Geml, Adrienn Geiger, Carla Mota Leal, Glodia Kgobe, Adrienn Mária Tóth, Szabolcs Villangó, Lili Mézes, Márk Czeglédi, György Lőrincz, Zsolt Zsófi

**Affiliations:** 1Food and Wine Research Institute, Research and Development Center, Eszterházy Károly Catholic University, Leányka u. 6, 3300 Eger, Hungary; molnar.anna@uni-eszterhazy.hu (A.M.); geiger.adrienn@uni-eszterhazy.hu (A.G.); mezes.lili@uni-eszterhazy.hu (L.M.); czegledi.mark@uni-eszterhazy.hu (M.C.); 2ELKH–EKKE Lendület Environmental Microbiome Research Group, Eszterházy Károly Catholic University, Leányka u. 6, 3300 Eger, Hungary; lmota.carla@gmail.com; 3Doctoral School of Environmental Sciences, Hungarian University of Agricultural and Life Sciences, Páter K. u. 1, 2100 Gödöllő, Hungary; kgobeglodia@gmail.com; 4Institute for Viticulture and Enology, Faculty of Natural Sciences, Eszterházy Károly Catholic University, Leányka u. 6, 3300 Eger, Hungary; toth.adrienn@uni-eszterhazy.hu (A.M.T.); villango.szabolcs@uni-eszterhazy.hu (S.V.); lorincz.gyorgy@uni-eszterhazy.hu (G.L.); zsofi.zsolt@uni-eszterhazy.hu (Z.Z.)

**Keywords:** endophytic fungi, gas exchange, microbiome, viticulture, *Vitis*, water potential

## Abstract

Improving our knowledge on biotic and abiotic factors that influence the composition of the grapevine mycobiome is of great agricultural significance, due to potential effects on plant health, productivity, and wine characteristics. Here, we assessed the influence of scion cultivar on the diversity and composition of fungal communities in the berries and leaves of three different cultivars. We generated DNA metabarcoding data, and statistically compared the richness, relative abundance, and composition of several functional groups of fungi among cultivars, which are partly explained by measured differences in chemical composition of leaves and berries and physiological traits of leaves. Fungal communities in leaves and berries show contrasting patterns among cultivars. The richness and relative abundance of fungal functional groups statistically differ among berry and leaf samples, but less so among cultivars. Community composition of the dominant functional groups of fungi, i.e., plant pathogens in leaves and saprotrophs in berries, differs significantly among cultivars. We also detect cultivar-level differences in the macro- and microelement content of the leaves, and in acidity and sugar concentration of berries. Our findings suggest that there appears to be a relatively diverse set of fungi that make up the grapevine mycobiome at the sampled terroir that spans several cultivars, and that both berry and leaf mycobiomes are likely influenced by the chemical characteristics of berries and leaves, e.g., pH and the availability of nutrients and simple carbohydrates. Finally, the correlation between fungal community composition and physiological variables in leaves is noteworthy, and merits further research to explore causality. Our findings offer novel insights into the microbial dynamics of grapevine considering plant chemistry and physiology, with implications for viticulture.

## 1. Introduction

Grapevine, one of the major crops grown principally for the production of table grapes and wines, is naturally colonized by a diverse array of microorganisms modulating plant health, growth, and crop yield and quality [1,2]. Fungi constitute the dominant component of the grapevine-associated microbiota, with a wide range of ecological roles [2,3]. The phyllosphere, i.e., the ephemeral above-ground green parts comprising the shoots, leaves, and reproductive structures of vascular plants, offers a variety of niches for fungal pathogens, commensal, litter, and wood saprotrophs, as well as mutualists that promote plant growth and stress tolerance [3]. The phyllosphere is a dynamic and harsh habitat for microbial colonizers, originating from the surrounding atmosphere [4], soil [5], and animal vectors [6], due to high levels of ultraviolet (UV) radiation, and rapid fluctuations in temperature and surface moisture [7]. Population sizes of the microbial inhabitants shift in distinct seasonal trends based on the growing season and, ultimately, leaf senescence [8].

In addition to environmental factors, biotic factors, e.g., host genotype, and management type may also shape the composition and diversity of the grapevine microbiome. Lately, culture-dependent and culture-independent methods were used to investigate changes in fungal communities of above-ground grapevine tissues among different types of plant protection, geographical locations, climatic conditions, seasons, and cultivars, although the grapevine phyllosphere is still less intensively studied than the rhizosphere [9,10,11,12,13,14,15,16,17,18,19]. For example, several studies found compositional differences among geographical locations in microbial communities associated with grapevine leaves [10], plant parts [20], and cultivars [21]. These results indicate that many biotic and abiotic factors influence the grapevine microbiome, although the results are contradictory in several cases. The functionality of phyllosphere fungi, and the environmental factors influencing it, are scarcely known, and the same is true for the relationships between leaf- and berry-associated fungi, and grapevine physiology and chemistry.

In this study, we characterized taxonomic composition and inferred functionality of phyllosphere fungal communities in three grapevine cultivars, *V. vinifera* cv. Furmint, cv. Kadarka, and cv. Syrah, grown in the same Grand Cru vineyard. As it was suggested that identity of the host plant, as well as chemical properties and physiology, can alter the microbial community structure [21], we focused on the question of whether the grapevine cultivar influences the fungal communities inhabiting the inner and external tissues of healthy (i.e., without visible symptoms of fungal infection) leaves and berries. To better understand the relationships between the fungal community and grapevine physiology and chemistry, we also tested for correlations between mycobiome composition and physiological parameters measured in situ, as well as macro- and microelement composition of leaf samples and the sugar concentration of the grape berries.

Specifically, our goals were to (1) characterize the genotypic richness and composition of various functional groups of fungi associated with leaves and berries of grapevine, (2) to test if the observed differences in richness or community composition are related to cultivar, and (3) to explore relationships between fungal community composition and grapevine leaf physiology and leaf and berry chemistry.

## 2. Results

### 2.1. Meteorological and Physiological Data

Weather parameters (mean annual temperature and annual precipitation) of the sampling year were 11.5 °C and 638.5 mm, respectively, which is slightly higher compared to the average (10 °C according to Köppen classification). None of the measured leaf gas exchange parameters (stomatal conductance, assimilation rate, and transpiration) differ significantly among grapevine cultivars. The determined assimilation rate (P_n_) values are the following: 5.2–8.8 mmol m^−2^ s^−1^ in Furmint, 7.2–8.8 mmol m^−2^ s^−1^ in Kadarka, and 6.0–7.4 mmol m^−2^ s^−1^ in Syrah. Values of stomatal conductance (g_s_) range from 87 mol m^−2^ s^−1^ to 178 mol m^−2^ s^−1^ in Furmint, from 73 mol m^−2^ s^−1^ to 181 mol m^−2^ s^−1^ in Kadarka, and from 53 mol m^−2^ s^−1^ to 103 mol m^−2^ s^−^^1^ in Syrah. Transpiration (E) ranges from 1.28 mol m^−2^ s^−1^ to 2.19 mol m^−2^ s^−1^ in Furmint, from 1.15 mol m^−2^ s^−1^ to 2.37 mol m^−2^ s^−1^ in Kadarka, and from 0.87 mol m^−2^ s^−1^ to 1.68 mol m^−2^ s^−1^ in Syrah. Pre-dawn water potential measurements indicate moderate water deficit (0.3–0.4 MPa), with no differences among cultivars [22,23].

### 2.2. Grapevine Mycobiome

We identified at the species or genus level a total of 568 fungal amplicon sequence variants (ASVs) in healthy berries, and 797 in healthy leaves, of three different cultivars of V. vinifera. The DNA sequences of fungal ASVs were deposited in GenBank (submissions ON864449–ON865305 for berries and ON865306–ON866491 for leaves). The genera with the highest number of ASVs in berries are *Aureobasidium*, *Alternaria*, and *Vishniacozyma* in Furmint and Kadarka, and *Aureobasidium*, *Alternaria*, and *Dioszegia* in Syrah (Figure 1). Based on observed incidence in all sampled leaves, three genera dominate in all cultivars: the plant pathogenic *Erysiphe* and *Alternaria*, and the litter saprotroph Cladosporium.

Berry fungal communities are dominated by generalist saprotrophs, which represent the only functional guild that is significantly more diverse in berries than in leaves, while plant pathogens and litter and wood saprotrophs have the highest ASV richness in leaves (Figure 2). Relative abundance values show similar patterns to those observed in richness, except that the read counts of plant pathogenic fungi are comparable in berries and leaves. In general, neither richness nor relative abundance of the functional groups differ significantly among cultivars in leaves or berries, with the only significant difference observed in read counts between Furmint and Kadarka leaves (Figure 2).

NMDS ordinations of fungal communities in grape berries and leaves (Figure 3). Results from the PerMANOVA indicate a significant effect of cultivar on fungal community structure in leaves (*p* = 0.0008), explaining 10.49% of compositional variance among all leaf samples, while the cultivar has a non-significant effect on fungal communities in berries. When beta diversity of leaf-associated fungi is partitioned into replacement and nestedness components, we observe that replacement accounts for most of the observed beta diversity, although nestedness is relatively high (Figure 4 and Figure 5) compared to what is observed in soil fungal communities [24]. In general, pairwise beta diversity measures are comparable within and between cultivars, with the exception that community turnover among samples within Syrah tend to be significantly lower than in Syrah–Furmint and Syrah–Kadarka pairs, particularly in berries (Figure 4). Linear regression analyses indicate significant positive relationships between pairwise differences in replacement and physiological parameters, and all three cultivars combined (Figure 5). We observe similar patterns when functional groups are analyzed separately, but correlations are only significant for plant pathogens and saprotrophs (data not shown). When data from different cultivars are analyzed independently, Furmint and Kadarka show the above-mentioned positive relationship between community turnover and differences in measured physiological parameters, albeit significant correlation is only observed for stomatal conductance in Furmint. Conversely, replacement in fungal communities in Syrah shows negative correlations with differences in transpiration rate and stomatal conductance (Figure 5). In Kadarka, none of the physiological parameters correlate significantly with replacement, and no correlation is detected for nestedness in any cultivar.

The indicator species analysis identified several characteristic fungal species associated with berries and leaves of each studied grapevine cultivar (Table 1). In berries, indicators for Furmint include species in the genera *Sporidiobolus*, *Hyphodontia*, *Rhodotorula*, *Sporobolomyces*, *Briansuttomyces*, and *Gibberella*. Numerous ASVs in the plant pathogenic fungal genera *Botryosphaeria* and *Alternariaster* are indicators for *Kadarka* berries; additionally we found *Pyrenochaetopsis* and *Vishniacozyma* as indicators. The lowest number of indicator ASVs are observed in Syrah berries, and belonged to the genus *Keissleriella*. Taxa significantly associated with Furmint leaves are identified mainly as *Aureobasidium*, *Sporobolomyces*, and *Dioszegia*, alongside with the genera *Microstroma* and *Ustilago*, while indicators for Kadarka include well-known plant pathogenic genera *Erysiphe*, *Comoclathris*, **Septoria**, and *Diplodia*. Syrah leaves are characterized by a wide range of indicators that, similarly to other cultivars, include *Aureobasidium*, *Erysiphe*, and *Ustilago* species.

### 2.3. Chemical Parameters of Leaves and Berries

We observe significant differences in several chemical parameters of the fully mature berries of the three studied cultivars, with Furmint and Syrah being the most different, and Kadarka generally showing intermediate values. The concentration of simple sugars, measured here as Brix index, is highest in Furmint, followed by Kadarka, with Syrah containing a significantly lower amount of sugar compared to both Hungarian cultivars. A similar pattern is observed for acidity, as indicated by the decrease in total acid content and the increase in pH from Furmint to Kadarka and then Syrah. Alpha-amino nitrogen content is highest in Syrah, with significantly lower values observed in Kadarka and in Furmint (Table 2). The measured concentrations of phosphorous (P), nitrogen (N), iron (Fe), calcium (Ca), and magnesium (Mg) in leaves are significantly different among the three cultivars, although we do not observe general trends as in the berry samples (Table 2). Kadarka leaves have the highest amount of P, Furmint the most Fe, and Syrah contains the highest concentrations of Ca, Mg, and N (Table 2).

## 3. Discussion

This paper provides novel insights into the compositional dynamics of fungal communities in grapevine leaves and grape berries in various cultivars. Specifically, this study is among the first to simultaneously compare leaf and berry mycobiome among scion cultivars, and to explore potential links between leaf mycobiome and physiology. The data presented here show that (1) different functional groups of fungi dominate leaf and berry communities, with plant pathogens and litter decomposers dominating in leaves, and generalist saprotrophs being the most diverse and abundant functional guild in berries; (2) there are significant differences among scion cultivars in community composition of several functional guilds of fungi and in chemical parameters of leaves and berries; and (3) we identify correlation between leaf mycobiome composition and leaf physiological activity, with possible implications for grapevine condition and productivity.

Our finding regarding the lack of significant effect of cultivar on richness and relative abundance of fungal functional groups in leaves and berries agrees with previous findings on grapevine-associated fungal communities [12,15], although our paper is the first to reveal differences among functional groups dominating berry vs. leaf samples. Our observation that differences in the mycobiome are greater among green, annual plant parts of the same grapevines than among cultivars, mirror the differences in fungal communities found in perennial woody parts of grapevine by Geiger et al. [19]. The dominance of plant pathogens and litter decomposers in leaves seems logical based on the life strategies of these fungi. Many of the plant pathogenic genera are among the most common leaf endophytes that can remain asymptomatic for relatively long periods (e.g., *Alternaria*, *Erysiphe*). Similarly, many litter decomposers (e.g., *Cladosporium*) are known to colonize green, asymptomatic leaves, and remain latent until leaf senescence, a strategy to dominate the senescing leaf and limit competition from secondary colonizers [25]. The high richness and abundance of generalist saprotrophs in berries can be explained by the fact that ripened grapes are rich in easily degradable simple carbohydrates, as opposed to leaves, where saprotrophs capable of degrading complex carbohydrates (e.g., cellulose) dominate.

The compositional differences of fungal communities among cultivars are driven by different functional groups in leaves and berries. The compositional differences are not particularly strong, and are significant only in the most diverse functional guild in the respective plant parts. In leaves, plant pathogenic fungi are the only functional guild that show significant cultivar effect, while saprotrophs contribute most of the compositional difference in berries (Figure 2). Singh et al. [14] detected significant cultivar effect on grapevine mycobiome composition, although, unlike in our study, the differences among cultivars were somewhat greater in berries than in leaves.

Host plant genotype is known to affect phyllosphere mycobiome in various agricultural crops [26], but the mechanisms behind this phenomenon are poorly known. The observed chemical differences among the three cultivars, notably N, P, Ca, and Mg content in leaves, and acidity and sugar content in berries, likely explain at least part of the compositional differences of the leaf and berry fungal communities among the cultivars. Unlike for physiological measurements, we could not directly test for correlation between mycobiome and chemical composition of leaf and berry samples, because, although sampled at the same time, not exactly the same berries and leaves were used for the mycobiome and chemical analyses, due to the destructive nature of these methods. Therefore, we can only hypothesize regarding the possible connections between the chemical and microbial composition of these plant parts. For example, the significant differences in berry pH among the cultivars may contribute to the compositional alterations in the fungal community, particularly that of generalist saprotrophs, which is the most diverse and abundant functional guild in berries (Figure 1 and Figure 2). pH is known to be an important factor influencing fungal community composition in general, e.g., by altering nutrient availability, with most supporting data originating from soil studies (e.g., [27,28,29,30]). Similarly, the fact that Syrah has significantly higher berry and leaf N content and leaf Ca and Mg content than the local varieties may explain some of the observed differences in mycobiome composition. As in the case of pH, N and P content is known to affect fungal communities strongly, as most terrestrial habitats are N- and/or P-limited, and are characterized by high C/N ratios [27,30]. Both N and P are vital for plant growth, and are taken up from the soil in inorganic forms, mostly via symbiotic microorganisms [31,32]. Despite previous reports on the positive relationship between N and P and the abundance of plant pathogenic fungi and/or disease severity in grasslands and agricultural systems [33,34], we did not find significant differences in the relative abundance of plant pathogens among cultivars, despite the pronounced differences in N and P content. It is important to note that plant N nutrition status could have contrasting effects on the development and severity of diseases, depending on the plant genotype, environment, and the strategy of the pathogen [35]. High N availability may result in greater disease severity, because of the increased green biomass that could create a more favorable microclimate for pathogens, as well as more N available for the growth of the pathogen itself [36,37]. However, favorable N status can also enhance plant defense [38]. Of the dominant plant pathogenic genera, only the powdery mildew genus *Erysiphe* shows higher relative abundance in Syrah and Kadarka, the cultivars with the greatest N and P content, respectively, than in Furmint, which has medium levels of both elements. The significance of iron as a micronutrient is crucial, as its complex mediates electron transfer during photosynthesis [39]. The role of Ca in shaping fungal communities, other than influencing pH, has been documented before in soil samples [27]. Calcium is a universal signaling molecule, and plays essential roles in a wide range of cellular processes of fungi, such as growth, reproduction, stress tolerance, and pathogen virulence [40]. Calcium is absorbed by plants in an ionic form, and is involved in maintaining the water balance in the cells [41]. Magnesium is a component of chlorophyll molecules and serves as an activator of several enzymes that catalyze carbohydrate metabolism. It is also involved in cellular pH and protein synthesis regulation [42]. Although the exact mechanisms are unknown, we hypothesize that the chemical differences among the cultivars possibly alter the competitive dynamics of leaf- and berry-associated fungi, thus, representing certain environmental filters for fungi with respect to establishment and persistence in the community. Targeted future studies, ideally spanning multiple vintages, are needed to investigate the causal relationships between plant chemical characteristics and phyllosphere fungal community dynamics.

Our finding that measured leaf physiological parameters correlates significantly with fungal community composition, which implies that some leaf-associated fungi may directly influence plant physiological processes. This conclusion is based on the following: (1) while we do not find physiological differences among cultivars, the more compositionally different leaf fungal communities are when all samples are considered, the greater the physiological differences we observed among leaves; (2) although fungal communities themselves are temporarily dynamic, the temporal variability of the measured physiological parameters is much greater, as changes can occur at the scale of minutes, which precludes the physiological parameters being the drivers of fungal community composition. We cannot prove the causal relationship between the leaf mycobiome and leaf physiology with the data at hand, and more studies are most certainly needed that focus on the role of leaf fungi on plant physiology. It is interesting to note that while physiological differences, particularly stomatal conductance, show positive relationships with the replacement component of fungal beta diversity in Furmint, the relationship is negative in Syrah. It appears that in the latter, not replacement, but nestedness, i.e., gain or loss of fungal species, and possibly the resulting loss of certain functions, could contribute to the physiological change. This is also supported by the positive relationship between differences in fungal richness and in physiological parameters in Syrah (data not shown). It is possible that the loss of certain fungal species, either due to fungicide treatment, competitive interactions, or to random drift, can result in altered physiological performance from the plant. Again, specific studies targeting these hypotheses are needed to reveal the mechanisms of any causal relationships between plant mycobiome and physiology.

## 4. Materials and Methods

### 4.1. Sampling Site

Sample collection took place on the south-facing slopes of the Nagy-Eged Hill, a historical Grand Cru terroir in the Eger wine region with favorable insolation and mesoclimate, and soils with neutral pH and moderate water-holding-capacity, developed on marine limestone [43]. To minimize the environmental effects, sampling was carried out in one particular vineyard, where the three cultivars were grown in close proximity, under identical environmental and management conditions. Furmint, a white variety, is autochthonous in the Carpathian Basin, and plays important role in the production of ‘Aszú’ wines [44]. Kadarka was a prevailing red variety of the Hungarian wine regions for centuries, originating from the Balkan Peninsula [45], and Syrah is a well-known Rhône Valley red cultivar, now planted worldwide [46]. Viticultural characteristics of these cultivars are shown in Table 3. Samples were collected on 9 September 2020, from three different parcels less than 200 m from each other, and located at the same elevations of 310–380 m above sea level (Furmint: 47.922753, 20.413823; Kadarka: 47.922269, 20.411916; Syrah: 47.922298, 20.412999). Continuously recorded temperature and precipitation data were obtained for the entire year of 2020 from the automatic weather station (Boreas Ltd., Érd, Hungary) installed in 2019 in the sampled vineyard. Degree days (above 10 °C) in the growth period (1 April–30 September) were calculated based on daily mean temperature values. All cultivars were grafted onto Fercal rootstock in 2008, and the vine spacing was 2 × 1 m (inter- and intra-row, respectively). The vines were cordon-trained with the same crop load (8 buds/plant). The same conventional plant protection management was used for all three cultivars on the same spraying days, including conventional fungicide, herbicide, and insecticide treatments, with additional cold-pressed orange oil adjuvant and powdery mildew, downy mildew, and grey rot-specific treatments. Chemicals used included herbicides: Chikara Duo (6.7 g/kg flazaszulfuron + 288 g/kg gliphosate), Pledge 50 WP (500 g/kg flumioxazin), Kabubki (26.5 g/L piraflufen-etil); insecticide Wakizasi (50 g/kg lambda-cyhalothrin); leaf fertilizer Im Plonvit Calcium Turbo (260 g CaO); effect enhancer Wetcit (fatty alcohol-etoxylate); and fungicides: Sercadis (300 g/L fluxapyroxad), Mildicut (25 g/L cyazofamid), Altima (500 g/L fluazinam), Teson (250 g/L tebuconazole), Dionys 80 WG (800 g/kg folpet), Cosavet DF (80% micronized Sulphur), Karathane Star (350 g/L meptyldinocap), Cymbal 45 WG (450 g/kg cymoxanil), and Chorus 50 WG (500 g/kg cyprodinil).

### 4.2. Leaf Gas-Exchange and Pre-Dawn Water Potential

From each cultivar, the following physiological measurements related to metabolic activity of 5 fully developed, asymptomatic leaves were taken on 9 September 2020: midday net CO_2_ assimilation rate (*P*_n_), stomatal conductance (*g*_s_), and transpiration rate (*E*) were determined. Measurements were carried out in the early afternoon (between 13:00–14:00, local time), when leaves were fully exposed to the sun, according to the local weather conditions and the south–north row orientation. A CIRAS-1 portable infrared gas analyzer (PP System, UK) (with leaf chamber type B) was used for the measurement of gas exchange parameters. All settings of the device met the specifications of the manual according to the leaf chamber used (flow rate 200 mls/min, RB 0.27 m^2^/s/mol, TR 0.14). Reference CO_2_ concentration was set to 410 ppm, and the photosynthetically active radiation (PAR) was above saturating light intensity during the measurements. All gas exchange records were taken on healthy, mature, and undamaged leaves fully exposed to the sun in five replicates per cultivar. All physiological measurements were taken within 1 h in all three cultivars, in order to obtain comparable data. In terms of light intensity (photosynthetically active radiation; PAR), vapour pressure deficit (VPD), and temperature (T), no difference was detectable in the experimental area during the sampling.

In order to assess soil water status of the sampling places, pre-dawn water potential was recorded with Scholander type pressure chamber [50]. Six undamaged healthy leaves were selected for each cultivar and the measurements were conducted between 2:00–3:00 h at night.

### 4.3. Sample Collection and Metagenomic DNA Extraction

Following the above-mentioned daytime physiological measurements, the same leaves were collected using sterile surgical gloves, and were placed in hermetic plastic bags. Berry samples were collected at cultivar-specific harvest times: on 22 September (Furmint), on 9 October (Kadarka), and on 20 October (Syrah). Samples were stored at −80 °C until further processing. After lyophilization, plant materials were disrupted and homogenized in a TissueLyser LT (QIAGEN, Hilden, Germany). Genomic DNA extraction was performed using NucleoSpin Plant II DNA Isolation Kit (MACHEREY-NAGEL, Düren, Germany), following the manufacturer’s instructions. ITS2 rDNA metabarcoding data were generated from all samples, using primers fITS7 [51] and ITS4 [52] appended with Illumina adaptors. Amplification and sequencing were performed on an Illumina NovaSeq at BaseClear (Leiden, The Netherlands), generating 250 base paired-end reads.

### 4.4. DNA Metabarcoding Data Analysis

Raw DNA sequences were processed with the *dada2* package [53], implemented in R v. 3.6.2 (R Development Core Team 2013, Vienna, Austria), designed to resolve fine-scale DNA sequence variation with improved elimination of artificial sequences. As *dada2* does not involve clustering sequences into OTUs, and is robust for removing spurious data, the output of unique ASVs captures both intra- and interspecific genetic variation of fungi found in the samples. This allows for the exploration of strain-level intraspecific differences. Raw sequences were truncated to 240 base pairs for forward and 200 for reverse reads, and were denoised, chimera-filtered, merged, and clustered into sequence variants. The maximum number of expected errors (maxEE) allowed in a read was 2. Taxonomic assignments of fungi were made with USEARCH v. 11 [54], based on the latest version (10 May 2021) of the UNITE database of reference sequences that represents all fungal species hypotheses (SHs) based on a dynamic delimitation [55]. We assigned fungal ASVs to putative functional guilds using the curated FungalTraits database [56], with the following modifications. Saprotrophic fungi that did not belong to litter and wood decomposers, i.e., nectar/sap saprotrophs, sooty molds, soil saprotrophs, and undefined saprotrophs, were treated as generalist saprotrophs primarily utilizing simple carbohydrates, hereafter referred to as “saprotrophs”. Also, “epiphytes” and “endophytes”, i.e., non-pathogenic leaf-associated fungi, were grouped into the category of “commensal” fungi.

### 4.5. Analytical Measurements

To determine the macro and micronutrients in grapevine leaves, 10 leaves were collected per cultivar on the above sampling day, and were subsequently dried and ground to powder. For each sample, 0.18 g of dried leaf powder was prepared for chemical analyses with 7 mL high-purity concentrated 68–70% of HNO_3_ (analytical pure grade, Fisher Chemical, Waltham, MA, USA) and 1 mL 30–32% H_2_O_2_ (ultrapure for trace metal analysis, Aristar, VWR Chemicals BDH.), and digested by Mars5 Microwave Digester System (CEM Corp., Matthews, NC, USA). After 20 min open digestion (reaction of volatile or easily oxidized compounds), the samples were digested with the plant tissue method (the original 400 W value was modified to 800 W for higher efficiency based on Sreenivasulu et al. [57]) in XP-1500 Plus-type vessels. Once cooled, the solution was diluted to 50 mL using ultrapure water. No further sample preparation was required, and no modifiers or ionization buffers were added. The sample preparation method was based on Dharmendra [58]. Two multi-element calibration standards were used, Fluka™ analytical standard for ICP 1–23 (100 mg/L each of element in 5% HNO_3_ matrix), and 7A (1000 mg/L K; 500 mg/L Si; 100 mg/L each of Al, B, Ba, Na, 50 mg/L Ag in 5% HNO_3_ matrix). All standards were prepared with 5% HNO_3_ (*v/v*) solution. The plastic rotation cubes, volumetric flask, and vessels were decontaminated with 10% HNO_3_ (*v/v*) for 24 h and, rinsed twice using 18.2 MW/cm deionized water before use [59]. The macro and microelement content of grapevine leaf samples was measured by microwave plasma–atomic emission spectrometry (MP–AES) (Model: 4200, Agilent Inc., Santa Clara, MA, USA) fitted with the nitrogen generator (Agilent 4107 type), applied double pass cyclonic spray chamber and OneNeb inert flow blurring nebulizer. Determination of carbon, nitrogen, and sulphur was carried out with an Elementar VarioMAXcube CNS analyzer (Elementar Analysensysteme GmbH, Langenselbold, Germany). A total of 100.00 mg samples of dried and shredded leaf samples were weighed into ceramic crucibles and analyzed. Calibration was carried out with 100.00 mg sulfanilic acid (>99.0% CNS standard grade, Sigma Aldrich, St. Louis, MO, USA).

In order to determine grape maturity stage, 15 clusters were collected from each cultivar. Berries were destemmed, crunched, and pressed. After pressing the berries, three replicates of must for each variety were provided for analytical measurements (50 mL per replicate). Sugar, total acid, and alpha-amino nitrogen contents, as well as the pH of the must, were determined with a WineScan™ FT 120 instrument (FOSS Analytical, Hillerød, Denmark), following the manufacturer’s protocol.

### 4.6. Statistical Analysis

All statistical analyses were performed in the R environment. ASVs with <10 reads in a given sample were excluded from that sample. In addition, ASVs that occurred in only one sample were deleted to minimize artifactual sequences. Normalization of the fungal community matrix (rarefaction) was performed by random subsampling to the smallest library size (353,226 reads for berry, 577,274 reads for leaf samples). The rarefied matrices contained 857 berry and 1186 leaf fungal ASVs that served as input for the subsequent analyses. To assess the effect of cultivar on general saprotrophs, wood saprotrophs, litter saprotrophs, plant pathogens, mycoparasites, and commensal fungi, ASV richness and relative abundance of these functional groups were statistically compared using ANOVA and Tukey’s HSD test, and were graphically presented as boxplots using the *ggplot2* R package [60]. Dissimilarities in composition among samples were visualized by non-metric multidimensional scaling analysis (NMDS) in the *vegan* R package [61], with Bray–Curtis distance measure on the Hellinger-transformed matrix. To estimate the amount of variation explained by the host cultivar, permutational multivariate analysis of variance (PerMANOVA) was carried out in the *vegan* R package. For leaf samples, we explored relationships between leaf mycobiome composition and measured leaf physiological parameters using linear regressions. For this purpose, we partitioned total beta diversity into replacement (i.e., turnover: the substitution of a species by a different one) and nestedness (where a poor community is the strict subset of a richer one) components. We used Sørensen dissimilarity as total beta diversity, and estimated the replacement (Simpson dissimilarity) and nestedness components on presence/absence data using the *betapart* R package [62]. We correlated the resulting pairwise Sørensen dissimilarity, Simpson dissimilarity, and nestedness values with pairwise differences of the measured physiological parameters. Physiological parameters were correlated with beta diversity measures individually, as well as a Euclidean distance matrix of the combined parameters standardized for mean and standard deviation. Indicator species analysis [63] was performed with the *multipatt* function in the *indicspecies* package [64], in order to identify characteristic fungal taxa for each cultivar.

## Figures and Tables

**Figure 1 plants-11-01924-f001:**
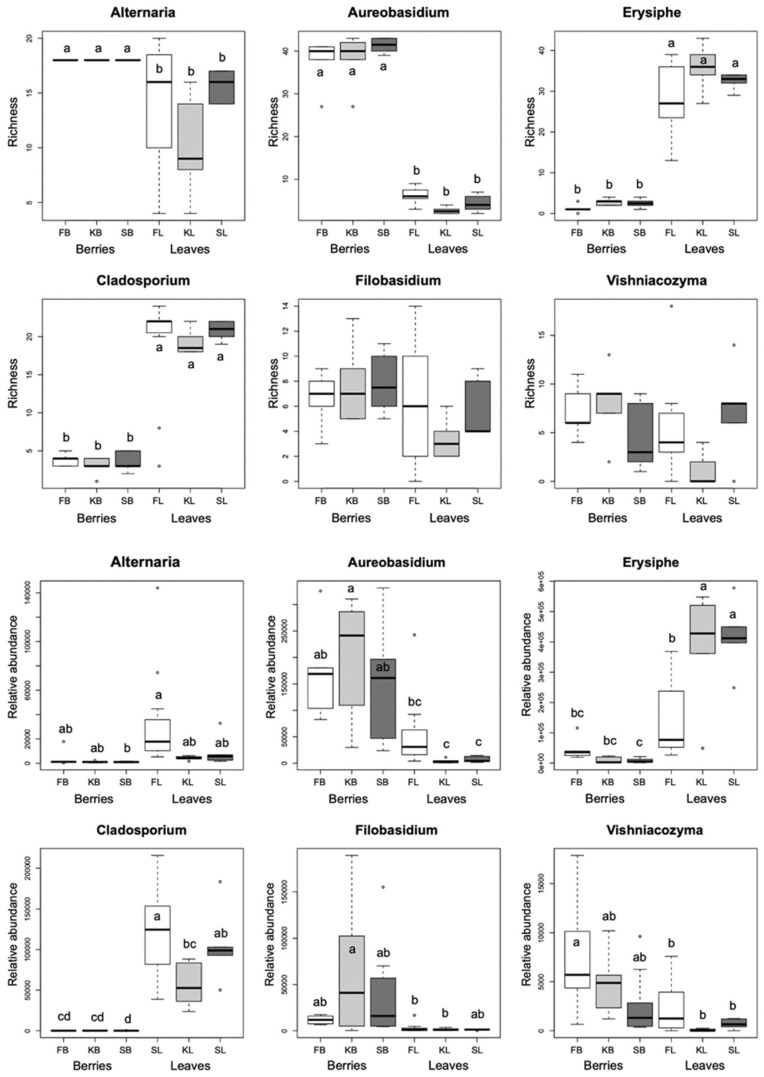
Boxplots showing the ASV richness and the rarefied read abundance of dominant fungal genera among all samples, based on the rarefied dataset. Means were compared using ANOVA and Tukey’s HSD tests, with letters denoting significant differences within each boxplot. Abbreviations: F—Furmint, K—Kadarka, S—Syrah, B—grape berry sample, L—leaf sample.

**Figure 2 plants-11-01924-f002:**
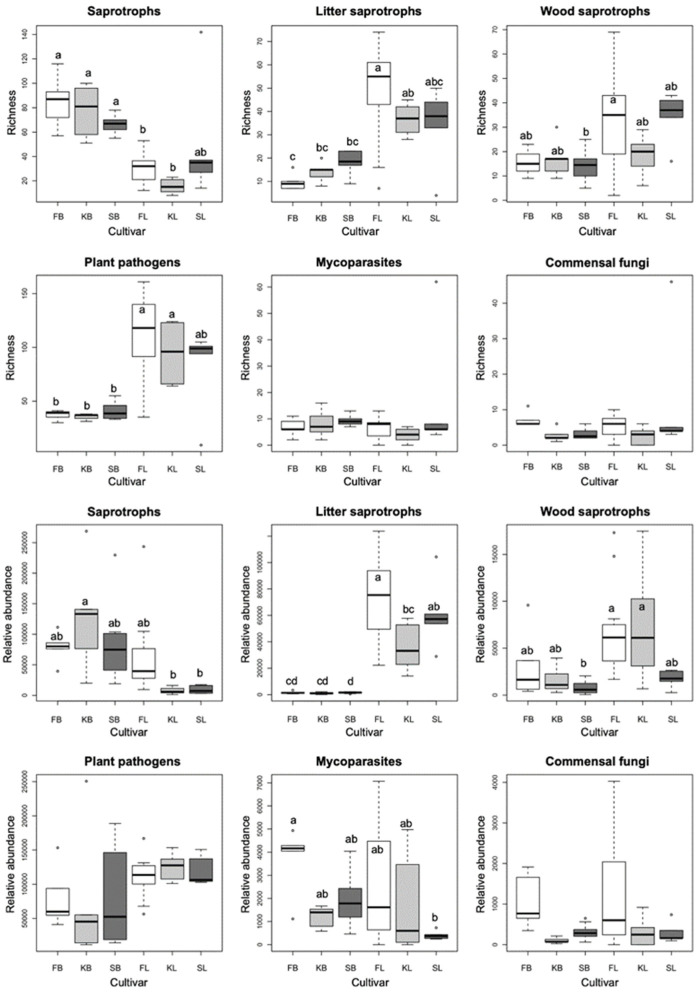
Boxplots showing the ASV richness and the rarefied read abundance of functional groups of fungi among all samples, based on the rarefied dataset. Means were compared using ANOVA and Tukey’s HSD tests, with letters denoting significant differences within each boxplot. Abbreviations: F—Furmint, K—Kadarka, S—Syrah, B—grape berry sample, L—leaf sample.

**Figure 3 plants-11-01924-f003:**
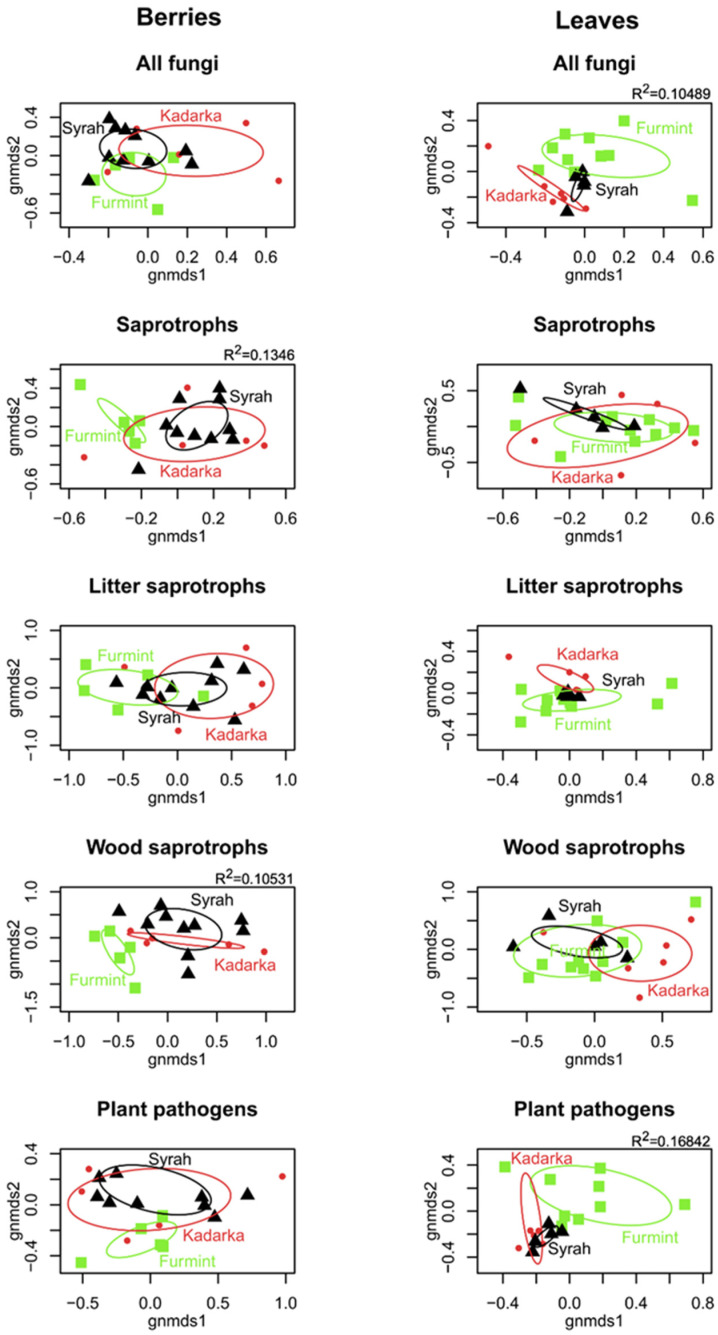
Non-metric multidimensional scaling (NMDS) ordination plots showing the differences in community composition of various functional groups of fungi among berry and leaf samples. The ordinations were based on a Bray–Curtis distance matrix generated from the Hellinger-transformed abundance table. Ellipses show the standard deviation of the compositional differences among samples from the same cultivar. R^2^ values indicate the explained variance of compositional differences among samples explained by the cultivar, based on PerMANOVA analyses (only statistically significant results are shown).

**Figure 4 plants-11-01924-f004:**
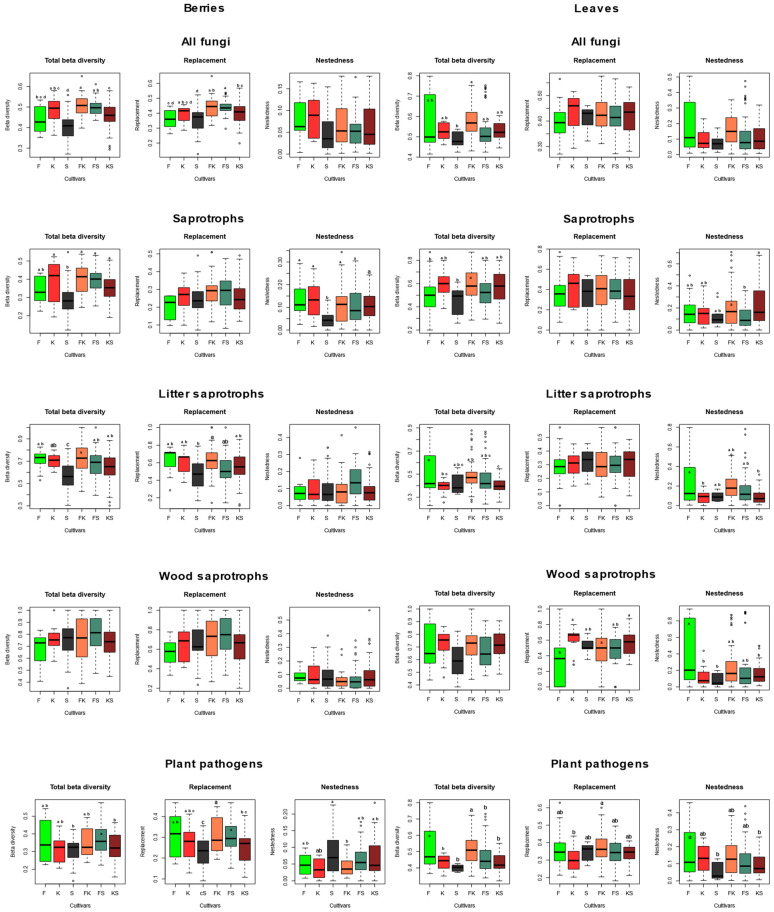
Boxplots showing the total beta diversity and its replacement and nestedness components in various functional groups of fungi, based on pairwise comparisons of samples within and between cultivars. Means were compared using ANOVA and Tukey’s HSD tests, with letters denoting significant differences within each boxplot. Abbreviations: F—Furmint, K—Kadarka, S—Syrah.

**Figure 5 plants-11-01924-f005:**
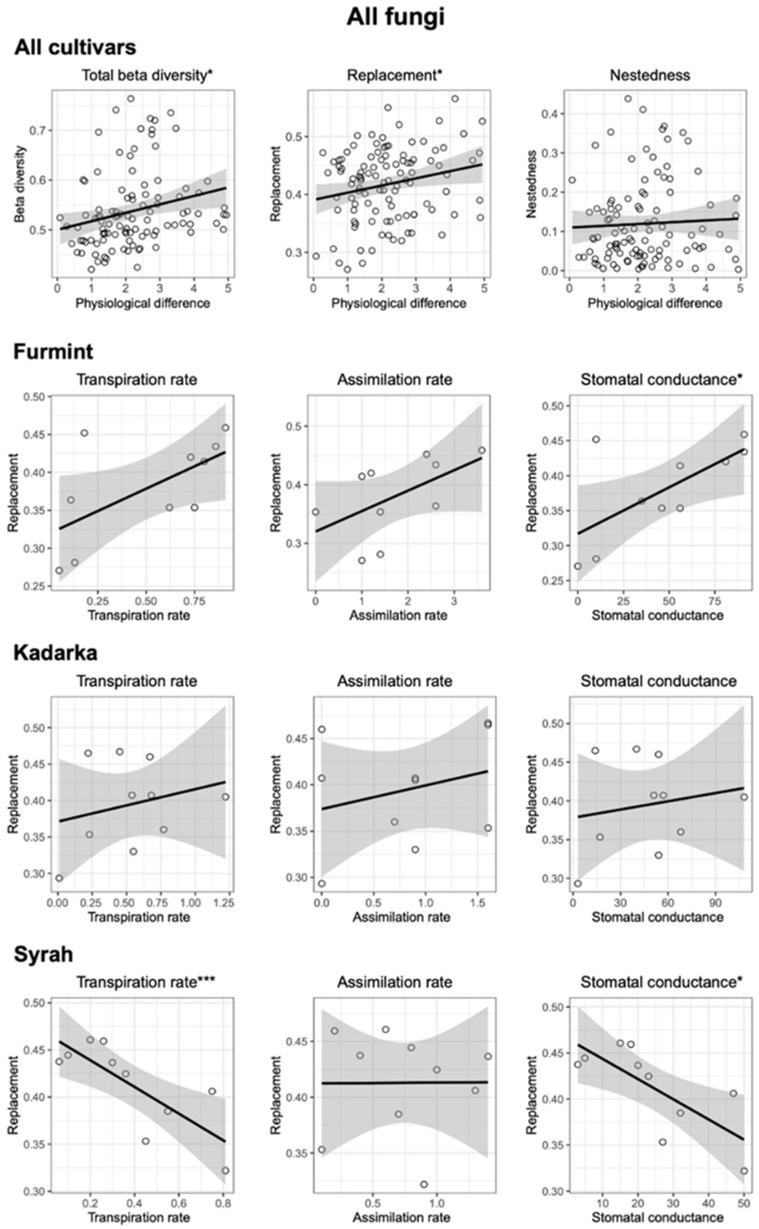
Correlation of total beta diversity and its replacement and nestedness components of the total fungal community, with pairwise differences in measured physiological characteristics factors in leaf samples. Asterisks indicate significance levels: *: *p* < 0.05 and ***: *p* < 0.001.

**Table 1 plants-11-01924-t001:** The list of indicator ASVs significantly associated with berries or leaves of a specific scion cultivar with the corresponding *p*-value, matching species hypothesis, ITS2 rDNA sequence similarity (%), taxonomic classification, and assigned functional guild of the most similar matching sequence in the UNITE+INSD dynamic species hypotheses database (version released on 10 May 2021).

ASV	*p*	SH	%	Matching Taxon	Class	Function
Berry						
Furmint						
ASV01064	0.016	SH1646443.08FU	95.4	*Agaricomycetes* sp.	Agaricomycetes	
ASV00648	0.015	SH1559713.08FU	99.6	*Lophiostomataceae* sp.	Dothideomycetes	
ASV00359	0.031	SH1189171.08FU	99.7	*Sporidiobolus* sp.	Microbotryomycetes	MP
ASV00229	0.025	SH1524265.08FU	99.6	*Pleosporales* sp.	Dothideomycetes	
ASV00731	0.023	SH1517815.08FU	99.7	*Hyphodontia quercina*	Agaricomycetes	WOOD
ASV00470	0.032	SH1185123.08FU	99.7	*Rhodotorula glutinis*	Microbotryomycetes	SAP
ASV00712	0.040	SH1575129.08FU	100	*Sporobolomyces ruberrimus*	Microbotryomycetes	MP
ASV00279	0.037	SH2232205.08FU	99.7	*Briansuttonomyces eucalypti*	Dothideomycetes	LITTER
ASV00441	0.037	SH1646414.08FU	98.8	*Agaricomycetes* sp.	Agaricomycetes	
ASV00355	0.045	SH2232205.08FU	100	*Briansuttonomyces eucalypti*	Dothideomycetes	LITTER
ASV00270	0.042	SH1610160.08FU	99.6	*Gibberella zeae*	Sordariomycetes	PPATH
Kadarka						
ASV00212	0.026	SH1635391.08FU	98.2	*Alternariaster helianthi*	Dothideomycetes	PPATH
ASV00061	0.017	SH1507512.08FU	98.3	*Botryosphaeria dothidea*	Dothideomycetes	PPATH
ASV00366	0.014	SH1188671.08FU	95.2	*Pyrenochaetopsis microspora*	Dothideomycetes	COM
ASV00027	0.024	SH1528219.08FU	98.9	*Vishniacozyma* sp.	Tremellomycetes	SAP
Syrah						
ASV00266	0.024	SH1564713.08FU	98	*Keissleriella taminensis*	Dothideomycetes	WOOD
Leaf						
Furmint						
ASV00142	0.013	SH1609774.08FU	99.6	*Dioszegia hungarica*	Tremellomycetes	LITTER
ASV00012	0.016	SH1515148.08FU	99.3	*Aureobasidium pullulans*	Dothideomycetes	SAP
ASV02203	0.046	SH1575129.08FU	100	*Sporobolomyces ruberrimus*	Microbotryomycetes	MP
ASV00908	0.022	SH1540544.08FU	96.3	*Microstroma bacarum*	Exobasidiomycetes	PPATH
ASV03652	0.038	SH1615186.08FU	96.7	*Ustilago crameri*	Ustilaginomycetes	PPATH
Kadarka						
ASV00352	0.001	SH1562822.08FU	100	*Erysiphe convolvuli*	Leotiomycetes	PPATH
ASV00051	0.011	SH1575708.08FU	96.9	Pleosporaceae sp.	Dothideomycetes	
ASV00257	0.016	SH1505878.08FU	96.8	*Comoclathris* sp.	Dothideomycetes	WOOD
ASV00454	0.013	SH1560734.08FU	100	Herpotrichiellaceae sp.	Eurotiomycetes	
ASV01091	0.028	SH2337384.08FU	99.6	*Septoria malagutii*	Dothideomycetes	PPATH
ASV00010	0.001	SH1171505.08FU	99	Ascomycota sp.	unidentified	
ASV00084	0.030	SH2131097.08FU	99.3	*Diplodia scrobiculata*	Dothideomycetes	PPATH
Syrah						
ASV00702	0.011	SH1565276.08FU	100	*Schizophyllum commune*	Agaricomycetes	WOOD
ASV00947	0.006	SH1155568.08FU	98.7	*Clonostachys rosea*	Sordariomycetes	WOOD
ASV00178	0.041	SH2176172.08FU	97.6	*Loratospora luzulae*	Dothideomycetes	LITTER
ASV00477	0.002	SH1246647.08FU	99.4	*Coprinellus xanthothrix*	Agaricomycetes	SAP
ASV00168	0.021	SH1599074.08FU	99.3	*Vuilleminia comedens*	Agaricomycetes	WOOD
ASV00521	0.025	SH2568050.08FU	98.4	*Erysiphe necator*	Leotiomycetes	PPATH
ASV02168	0.013	SH1574527.08FU	100	*Bulleromyces albus*	Tremellomycetes	SAP
ASV00847	0.039	SH1515148.08FU	99	*Aureobasidium pullulans*	Dothideomycetes	SAP
ASV04003	0.043	SH1555460.08FU	99	*Microdochium seminicola*	Sordariomycetes	PPATH
ASV02439	0.038	SH1643363.08FU	99.6	*Muriphaeosphaeria viburni*	Dothideomycetes	PPATH
ASV04868	0.043	SH1572518.08FU	99.7	*Melampsora laricis-populina*	Pucciniomycetes	PPATH
ASV01105	0.039	SH1539608.08FU	98.5	*Myrmecridium phragmitis*	Sordariomycetes	SAP
ASV01832	0.044	SH1542846.08FU	99.7	*Massaria anomia*	Dothideomycetes	WOOD
ASV01602	0.045	SH1509408.08FU	99	*Ustilago nunavutica*	Ustilaginomycetes	PPATH

Abbreviations for functional guilds: COM = commensalist, LITTER = litter decomposer, MP = mycoparasite, PPATH = plant pathogen, SAP = generalist saprotroph, WOOD = wood decomposer.

**Table 2 plants-11-01924-t002:** Chemical parameters of leaf and berry samples from the three investigated grapevine cultivars: Furmint (F), Kadarka (K), and Syrah (S).

	Furmint	Kadarka	Syrah
**Leaves**			
P **	1627.80 ± 151.84 ^b^	1923.57 ± 84.79 ^a^	1407.17 ± 75.19 ^b^
Fe *	76.47 ± 11.42 ^a^	65.77 ± 3.20 ^ab^	53.43 ± 1.25 ^b^
Zn	151.43 ± 9.49	138.50 ± 18.79	132.47 ± 1.33
Ca **	24,795.57 ± 1650.34 ^b^	23,770.10 ± 2531.14 ^b^	30,613.27 ± 967.30 ^a^
Cu	11.20 ± 4.62	9.57 ± 4.61	5.77 ± 1.03
B	115.20 ± 14.13	93.67 ± 19.61	94.33 ± 4.34
K	6897.20 ± 321.18	6903.80 ± 614.66	6083.83 ± 466.27
Mn	132.10 ± 26.69	163.73 ± 3.01	163.50 ± 5.09
Mg **	4584.60 ± 176.31 ^b^	4110.17 ± 99.94 ^b^	5604.50 ± 597.11 ^a^
Al	162.20 ± 97.33	93.40 ± 20.20	122.13 ± 1.91
Na	975.07 ± 45.35	1003.90 ± 89.86	1075.10 ± 18.53
N **	1.94 ± 0.07 ^b^	1.98 ± 0.08 ^b^	2.34 ± 0.01 ^a^
C	42.78 ± 0.53	42.38 ± 0.45	42.94 ± 0.33
S	0.19 ± 0.02	0.27 ± 0.06	0.19 ± 0.02
**Berries**			
Brix index ***	24.50 ± 0.00 ^a^	23.53 ± 0.45 ^b^	21.00 ± 0.00 ^c^
Total acid ***	7.33 ± 0.06 ^a^	4.57 ± 0.12 ^b^	4.07 ± 0.06 ^c^
Malic acid	2.63 ± 0.90	3.37 ± 0.02	3.46 ± 0.01
Alpha-amino nitrogen **	117.00 ± 2.65 ^b^	125.67 ± 4.93 ^ab^	133.33 ± 3.51 ^a^
pH ***	3.15 ± 0.01 ^c^	3.37 ± 0.02 ^b^	3.46 ± 0.01 ^a^

Macro- and micro-nutrient contents in leaves were determined with MP-AES and CNS, while sugar, total acid, and alpha-amino nitrogen contents, as well as pH, of berries were measured with a WineScan™ FT. Values are given as mean ± standard deviation. Significance was determined using one-way ANOVA (* = significant at *p* < 0.05; ** = significant at *p* < 0.1, *** = significant at *p* < 0.001). Values are expressed in ppm, in % dry weight for N, C, and S, and in g/L and in mg/L for acid concentrations. Within each row, letters denote significant differences based on Tukey’s HSD test.

**Table 3 plants-11-01924-t003:** Viticultural characteristics of the investigated grapevine varieties [47,48,49].

Cultivar	Furmint	Kadarka	Syrah
Origin	*convar. pontica* *subconvar. balcanica*	*convar. pontica* *subconvar. balcanica*	*convar. occidentalis* *subconvar. gallica*
Time of bud burst (OIV 301) ^a^	medium	late	early/medium
Time of full bloom (OIV 302)	late/very late	late	medium/late
Time of the onset of berry ripening (veraison (OIV 303)	medium	late	medium
Time of full physiological maturity of berry (OIV 304)	medium	late/very late	medium
Bunch: density (OIV 204)	dense	dense/very dense	medium/dense
Bunch: single bunch weight (OIV 502)	very low/low	medium/high	medium
Berry: single berry weight (OIV 503)	low	medium/high	low
Yield per m^2^ (OIV 504)	low/medium	high	high

^a^ based on standard grapevine descriptors of the Organisation Internationale de la Vigne et du Vin (OIV).

## Data Availability

DNA sequences of fungal ASVs generated in this study were deposited in GenBank (ON864449–ON865305 for berries and ON865306–ON866491 for leaves).

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
