# Peer review of "Exploring Relationships among Grapevine Chemical and Physiological Parameters and Leaf and Berry Mycobiome Composition"

_plants, 2022, doi:10.3390/plants11151924_

Round 1

Reviewer 1 Report

The manuscript of Molnár and colleagues reports the difference found in leaves and berries of three grapevine cultivars with respect to their physio-chemical features and microbiomes. The item sounds intriguing but aside from the huge work and computational elaborations some concerns were raised. (1) In the introduction the authors evidenced a broad variability in phyllosphere microbiomes tied to biotic and abiotic factors, but they don’t emphasize the importance of the phyllosphere’s microbiome (PM). I mean, how can you justify your choice in studying the phyllosphere instead of the rhizosphere, against the present trend? How this study could improve the common knowledge of PM? What do you expect? This is a crucial point to be improved because it could justify the study itself and publication as well. The highlight in the abstract (..agricultural significance due to potential effects on plant health, productivity, and wine characteristic) is not enough and should be consistently developed with relative references. (2) In spite of 568 fungal amplicons obtained by HTS from berries and 797 from leaves is not clear what’s the ratio of this choice operated in the study design…why the authors focused on a few culturable genera…Yes, are the major genera but what do you expect to note working at genus instead of at species level? Why produce so huge data if are used only a part (what’s the percentage covered?). (3) Less is better. Too many figures/graphs mainly unreadable (too little). The authors should synthesize relevant results. What is not so relevant should be reported as supplementary material. This is of utmost importance because (as the journal’s template suggests) the results should be understandable by reading the title and the introduction only. The figures and their captions should be conceived to guide the reader in understanding. As they are, are confusing and lose significance as a decoration on wallpaper. And that shouldn’t happen after so many efforts. (4) The discussion should be reworked. Being the first doing something is not necessarily relevant, so that doesn’t sound good tell and repeat it before to prove the relevance of the results. As usual, it should have a brief introduction on the items the authors wanted to deepen (maybe those highlighted in the abstract) and then discuss the achieved results. L207-213 are highlights, not discussions. As it is, the discussion is vague and too general.

Some additional comments

Due to the complexity of the analysis performed is of the utmost importance to prepare a synopsis/scheme of the study. as to easily understand the study design and results.

The title doesn’t fit the manus contents and in my opinion, should be modified. I mean, both results and M&M give more importance to the microbiome than physio-chemical analysis. That’s clear even considering the order in which the items have been reported. Moreover, there is no trace of the NMDR elaboration based on ecological traits and the purpose of the study.

The sequence data are not available.

L38-39 From an economic perspective, the perennial plant grapevine constitutes one of the major crops, grown principally for fruit and wine production [1]. That’s true but it is reductive. Grapevine accompanied humankind since its raising. There are documented signs several thousand years before the present era. Please improve the sentence.

L42… range of ecological roles. Add the reference

The phyllosphere, comprising of the surface and interior of …please improve this part by giving a sharp definition

L46 above-ground part please modify to the aerial part

L55 citations should be reported just before punctuation marks. Please check this item along with the text (other cases a ref51 and 52)

L59-62 remove “both” and before besides change “,” to “;” or “.”

L62-64 Please change to “These results indicate that many biotic and abiotic factors influence the grapevine microbiome, although the results are contradictory in several cases”

L65s the possible differences- Considering L57-63, it is certain more than a mere possibility

L97 V. vinifera - italics

Figures are too little, I mean the huge nr of diagram reported decreased readability. Less it’s better.

L98 Based on presence/absence data - What do you mean? Please explain

L99 Genera should be in italics please add a table/figure reference where this data can be confirmed.

L100-101 When assessing the genera represented in the highest number in leaves- that’s wordy, please rework this part. Maybe something like: The most represented genera on leaves were…or three genera dominated in number in all cultivars…could it be better

L102-103 the plant pathogenic Erysiphe and Alternaria and the litter saprotroph Cladosporium. (a) This sentence seems to contrast with the previous one where Cladosporium wasn’t mentioned. (b) here you are talking about the detection frequency, so the mention of their ecological trait is not relevant because reported in the results section. (c) the traits reported are like the iceberg top. Alternaria genus is composed mostly of saprobic or plant pathogen species but could be opportunistic for humans being also quite frequent in indoor environments (see sick building syndrome/wallpaper etc). Cladosporium counts hundreds of species with saprobic, plant-pathogen, and endophytic traits with the ability to occupy even extreme niches as salterns or sites contaminated by radioactivity.

L105-107 please give the names and not their traits

L203 dynamics? Dynamics presuppose a change of mycobiome in relation to time. For example, the mycobiome is measured in leaves (same species/same plant) during different periods of the year. But it is not the case.

L204-206 this sentence is not relevant.

L226-230 The high richness and abundance of generalist saprotrophs (please give examples) in berries can be explained by the fact that ripe grapes are rich in easily degradable simple carbohydrates as opposed to leaves, where saprotrophs are capable of degrading complex carbohydrates (e.g., cellulose) dominate ..this last sentence sounds bad because fungi able to degrade cellulose are also able to use simple sugars and mostly prefer them for fast growth and spreading.

L333-336 the fungicides/herbicides/insecticides should be cited. This is important to justify for example the absence of some common species. Moreover, a treatment considered conventional /common in a country it couldn’t be in another one.

L341 CO2, 2 should be subscript the same for the f HNO3 formula (L390) and H2O2 (L391) please check this item along with the text

L357 were placed hermetic plastic bags - missing in

L358 for DNA work. the present expression doesn’t sound like English. This information could be skipped because already present in the title.

L363 please explain why you choose ITS2 instead of ITS1 for metabarcoding analysis

L379-385 It has been explained the way to group fungal traits, but not why the fungal traits were considered for the analysis. Why they were considered for?

L413-418 epicuticular wax content it’s basic for berries protection and yeast content / wine flavour. Please explain (in the text) the reason for your choices (i.e. exclusion of wax measurement and the inclusion of the other parameters)

Author Response

Please, see our responses to Reviewer 1 in the attachment.

Reviewer 2 Report

The study showed information about the correlation between the chemical and physiological characteristic of three grapevines and the leaf and berry mycobiome composition. The aims of the work are well defined, and the methodology used in the assay responds in good form to the objectives. The assay is very interesting and offer novel insights in the microbiome communities associated to grapevines, but there is some issues that need revision after acceptance.

Material and methods section

Lines 322. Further information about the type of soil is necessary to shows the similarity said by the authors with parameters such as texture, fertility, soil deep. Please clarify.

Results section

Lines 97, 99. Scientific names in cursive letter

Discussion section

Lines 207-210. Is interesting that the authors in the entire text never make a link between the disease management in field and the type of fungal families founded in leaves and berries. In lines 331-334 there is many treatments used in the field such as fungicides, pesticides, herbicides and specific treatment for important fungal disease such as Botrytis that in certain form could modulate the fungal families encountered in the results. Maybe a phrase that take into account this comment is necessary.

Moreover, is very intriguing that there is no fungal belong to the non-Saccharomyces species. Maybe this is not the objective of the work but is an opportunity to this area to show the diversity of species in this important world viticulture zone. Perhaps this should also be made more explicit in the text and the authors could make the link between the species founded and the future wine characteristics only if the wineries use fermentation with indigenous yeasts, especially in the wines made with local varieties.

Lines 285-286. Although the assay was executed only in one season, a phrase is necessary to say that the use of two or more seasons could explain the results observed in only one season.

Lines 297. The author in paragraph says that leaf physiological parameters correlated significantly with fungal composition, but in lines 297, the authors says “We cannot prove the causal relationship between the leaf mycobiome and leaf physiology with the data at hand…” In my opinion, this phrase perhaps reflect that the physiological parameters are not so important, that is, if this information is removed, the research would lose weight? I think not, but it is something that the authors must decide.

Author Response

Please, see our responses to Reviewer 2 in the attachment.

Reviewer 3 Report

Thank you for submitting this manuscript for consideration. I enjoyed your efforts and findings. Please see my comments below. 

Line 39 - 40: 'grapevine is' should be changed to 'grapevines are'. 

Line 66: V. vinifera L. cv. Furmint

Line 97: V. vinifera should be italicized. Also, in this section shouldn't the fungal names also be italicized and full scientific names used? For example Erysiphe necator

Figure 1, 2 - Difficult to read, seems fuzzy. N = ?

Figure 3 - Red and Green can be difficult for some readers. Consider changing the color scheme. 

Table 2 - I would recommend making the Tukey letter full sized so they are easier to see. Usually superscripts are reserved for notes. 

Discussion - I would recommend doing away with 'our' 'we' etc. Instead consider 'Specifically, this study study is among....'  (Line 204 - 205. 

The beginning of this discussion is very similar to the introduction. Address how repetitive it reads. 

Section 4.2 - What were your device settings for CO2, flow rate etc? Please add more detail here. 

Section 4.6 - Version number of R? 

Line 456 - Typo. You have an extra ',' and some capitalization issues. 

Author Response

Please see our responses to Reviewer 3 in the attachment.

Round 2

Reviewer 1 Report

The manuscript has been revised by the authors, but the result does not meet the expectations and basic standards for publication in a Q1 and 4.658 IF journal. Many of the concerns raised before are still pending, and from the responses given, I have the impression that the authors are not familiar with the basic rules and aims of scientific communication and politeness. Science should be inclusive and able to involve/ catch the broader readership sharing information. If the information is understood by the authors only or by a few people, it is not effective communication. So, If a reviewer asks for an explanation or raises a question, it means that something is missing or not as clear in the text as believed by the authors. So that the authors are asked to improve the text and report the line and the sentence added and/or modified in the responses list. No private lectures are required because it’s possible that the reviewer already knows the answer but, as above, wants to push the authors to write it for the readers’ benefit. In a world with great accessibility to all kinds of data, constantly in a hurry and where the best result must be achieved in the shortest time possible is not realistic for example, to pretend from a reader the patience to enlarge each of your sub-graphs (30 in figure 4) to catch/study its meaning. The majority of readers will skip your work...quite bad being authors and journal reputations also measured by citations.

Regards